# HiViT: A Simpler and More Efficient Design of Hierarchical Vision Transformer

**Xiaosong Zhang**[1*]     **Yunjie Tian**[1*]     **Lingxi Xie**[2]     **Wei Huang**[1]     **Qi Dai**

**Qixiang Ye**[1,3†]                    **Qi Tian**[2]

[1]UCAS            [2]Huawei Inc.            [3]Peng Cheng Lab
{zhangxiaosong18, tianyunjie19, huangwei19}@mails.ucas.ac.cn
{198808xc, daiqi1989}@gmail.com qxye@ucas.ac.cn tian.qi1@huawei.com

## Abstract

There has been a debate on the choice of plain *vs.* hierarchical vision transformers, where researchers often believe that the former (*e.g.*, ViT) has a simpler design but the latter (*e.g.*, Swin) enjoys higher recognition accuracy. Recently, the emerge of masked image modeling (MIM), a self-supervised pre-training method, raised a new challenge to vision transformers in terms of flexibility, *i.e.*, part of image patches or tokens are to be discarded, which seems to claim the advantages of plain vision transformers. In this paper, we delve deep into the comparison between ViT and Swin, revealing that (i) the performance gain of Swin is mainly brought by a deepened backbone and relative positional encoding, (ii) the hierarchical design of Swin can be simplified into hierarchical patch embedding (proposed in this work), and (iii) other designs such as shifted-window attentions can be removed. By removing the unnecessary operations, we come up with a new architecture named **HiViT** (short for hierarchical ViT), which is simpler and more efficient than Swin yet further improves its performance on fully-supervised and self-supervised visual representation learning. In particular, after pre-trained using masked autoencoder (MAE) on ImageNet-1K, HiViT-B reports a $84.6\%$ accuracy on ImageNet-1K classification, a $53.3\%$ box AP on COCO detection, and a $52.8\%$ mIoU on ADE20K segmentation, significantly surpassing the baseline. Code is available at <https://github.com/zhangxiaosong18/hivit>.

## 1 Introduction

Deep neural networks (LeCun et al., 2015) have advanced the research fields of computer vision, natural language processing, *etc.*, in the past decade. Since 2020, the computer vision community has adapted the transformer module from natural language processing (Vaswani et al., 2017; Devlin et al., 2019) to visual recognition, leading to a large family of vision transformers (Dosovitskiy et al., 2021; Liu et al., 2021; Wang et al., 2021; Zhou et al., 2021; Dai et al., 2021; Li et al., 2021a) that replaced the dominance of convolutional neural networks (Krizhevsky et al., 2012; He et al., 2016; Tan & Le, 2019). They have the ability of formulating long-range feature dependencies, which naturally benefits visual recognition especially when long-range relationship is important.

There are mainly two families of vision transformers, namely, the plain vision transformers (Dosovitskiy et al., 2021; Touvron et al., 2021) and the hierarchical vision transformers (Liu et al., 2021; Wang et al., 2021; Dong et al., 2021a; Chen et al., 2021a), differing from each other in whether multi-resolution features are used. Intuitively, visual recognition requires hierarchical information and the hierarchical vision transformers indeed show superior performance. However, the hierarchical vision transformers have introduced complicated and asymmetric operations, *e.g.*, Swin (Liu et al., 2021) used regional self-attentions with shifted windows, hence, they encounter difficulties

---

[*]Equal Contribution.
[†]Corresponding Author.

when the tokens need to be flexibly manipulated. A typical example lies in masked image modeling (MIM), a recent methodology of visual pre-raining (Bao et al., 2021; He et al., 2021; Xie et al., 2021b), in which a random subset of image patches are masked from input and the model learns by reconstructing the masked contents. In such a circumstance, the plain transformers (*e.g.*, ViT) can directly discard the masked tokens, while the hierarchical vision transformers (*e.g.*, Swin) must feed the entire image (with the masked patches filled with dummy contents) into the encoder (Xie et al., 2021b), slowing down the training procedure and contaminating the original data distribution.

This paper tries to answer the following question: is it possible to design an alternative vision transformer that enjoys both the flexibility of plain models and the representation ability of hierarchical models? We start with ViT and Swin, the most popular plain and hierarchical models. We design a path that connects them, with each step only changing a single design factor. The modifications include (a) increasing network depth, (b) adding relative positional encoding, (c) adding hierarchical patch embedding, (c') adding shifted-window attentions (an alternative to (c)), and (d) adding the stage 4[1]. We find that (a)(b)(c) are the main factors that contribute to visual recognition, while (c') shall be replaced by (c) and (d) can be discarded. In particular, the window attentions were designed to reduce the computation of self-attentions in the high-resolution (*i.e.*, low-level) feature maps, but we find that, under a sufficient network depth (*e.g.*, Swin-B used 24 transformer blocks), the low-level self-attentions only have marginal contribution and can be removed.

Based on the analysis, we present a hierarchical version of ViT named **HiViT**. Following (a)(b)(c) discussed above, the modification beyond the original ViT is minimal. At the base level, the architecture has 24 transformer blocks (the number of channels is reduced) where the first 4 appear as hierarchical patch embedding that replaces the plain counterpart and the others are equipped with relative positional encoding – one needs only a few lines of code to replace ViT with HiViT.

The superiority of HiViT is validated using two sets of experiments. We first perform fully-supervised image classification on ImageNet-1K (Deng et al., 2009), where HiViT enjoys consistent accuracy gains over both ViT and Swin, *e.g.*, at the base level, HiViT-B reports a 83.8% top-1 accuracy, which is +2.0% over ViT-B and +0.3% over Swin-B, and the number of learnable parameters is about $1/4$ fewer than both competitors. Continuing to MIM, the advantages of HiViT persist. Table 1 shows the comparison on model size, training

Table 1: Compared to ViT and Swin, HiViT is faster in pre-training, needs fewer parameters, and achieves higher accuracy. All numbers in % are reported by pre-training the model using MIM (ViT-B and HiViT-B by MAE and Swin-B by SimMIM) and fine-tuning it to the downstream data. Please refer to experiments for detailed descriptions.

| Architecture | ViT-B | Swin-B | HiViT-B |
|---|---|---|---|
| Params (M) | 86.6 | 88.0 | 66.4 |
| FLOPs (G) | 17.5 | 15.4 | 15.9 |
| Pre-training time per epoch (mins) | 8.8 | 14.3 | 10.3 |
| ImageNet-1K acc. (%) | 83.6 | 84.0 | 84.6 |
| COCO AP$^{box}$ (%) | 51.2 | 52.3 | 53.3 |
| ADE20K mIoU (%) | 48.1 | 52.8 | 52.8 |

speed, and recognition accuracy. Under the MAE framework (He et al., 2021), with 1600 epochs of pre-training and 100 epochs of fine-tuning, HiViT-B reports a 84.6% top-1 accuracy on ImageNet-1K, which is +1.0% over ViT-B (trained with MAE) and +0.6% over Swin-B (trained with Sim-MIM (Xie et al., 2021b)). More importantly, HiViT enjoys the efficient implementation that discards all masked patches (or tokens) at the input stage, and hence the training procedure is as simple and efficient as applying MAE on ViT. The pre-trained models also show advantages on downstream tasks, including linear probing (a 71.3% top-1 accuracy on ImageNet-1K), semantic segmentation (52.8% mIoU on ADE20K (Zhou et al., 2017)), and object detection and instance segmentation (a 53.3% box AP and a 47.0% mask AP) on COCO (Lin et al., 2014a) under the $3\times$ training schedule.

Overall, the core contribution of this paper is HiViT, a hierarchical vision transformer architecture that is off-the-shelf for a wide range of vision tasks. In particular, with MIM being a generalized paradigm for self-supervised visual representation learning, HiViT has the potential of being directly plugged into many existing algorithms to improve their effectiveness and efficiency.

---

[1]Stage refers to components processing the same resolution in hierarchical models. In this paper, stage 1, 2, 3, and 4 respectively refer to components processing $56^2$, $28^2$, $14^2$, and $7^2$ resolutions in image classification.

## 2 RELATED WORK

**Vision transformers** (Dosovitskiy et al., 2021) were adapted from the natural language processing (NLP) transformers (Vaswani et al., 2017; Devlin et al., 2019), opening a direction of designing vision models with weak induction bias (Tolstikhin et al., 2021). Early vision transformers (Dosovitskiy et al., 2021) mainly adopted the plain configuration, and efficient training methods are strongly required (Touvron et al., 2021). To cater for vision-friendly priors, Swin Transformer (Liu et al., 2021) proposed a hierarchical architecture that contains multi-level feature maps and validated good performance in many vision tasks. Since then, various efforts emerged in improving hierarchical vision transformers, including borrowing design experiences from CNNs (Wang et al., 2021; Wu et al., 2021; Vaswani et al., 2021), adjusting the design of self-attention geometry (Dong et al., 2021a; Yang et al., 2021), and integrating convolution and transformer modules (Srinivas et al., 2021; Gao et al., 2021; Dai et al., 2021; Peng et al., 2021; Xiao et al., 2021; Guo et al., 2022; Ali et al., 2021; Pan et al., 2022), *etc*.

In the context of computer vision, **self-supervised pre-training** aims to learn compact visual representations from unlabeled data. The key is to design a pretext task that sets a natural constraint for the target model to achieve by tuning its weights. Existing pretext tasks are roughly partitioned into three categories, namely, geometry-based proxies that were built upon the spatial relationship of image contents (Wei et al., 2019; Noroozi & Favaro, 2016; Gidaris et al., 2018), contrast-based proxies that assumed that different views of an image shall produce related visual features (He et al., 2020; Chen et al., 2020; Grill et al., 2020; Caron et al., 2021; 2020; Xie et al., 2021a; Tian et al., 2021), and generation-based proxies that required visual representations to be capable of recovering the original image contents (Zhang et al., 2016; Pathak et al., 2016; He et al., 2021; Bao et al., 2021; Tian et al., 2022). After the self-supervised learning (*a.k.a.* pre-training) stage, the target model is often evaluated by fine-tuning in downstream recognition tasks – the popular examples include image classification (Deng et al., 2009), semantic segmentation (Zhou et al., 2017), object detection and instance segmentation (Lin et al., 2014b), *etc*.

We are interested in a particular visual pre-training method named **masked image modeling** (MIM) (Bao et al., 2021; He et al., 2021). The flowchart is straightforward: some image patches (corresponding to tokens) are discarded, the target model receives the incomplete input and the goal is to recover the discarded patches. MIM is strongly related to the masked language modeling (MLM) task in NLP. BEiT (Bao et al., 2021) transferred the task to computer vision by masking the image patches and recovering the tokens produced by a pre-trained model (knwon as the tokenizer). MAE (He et al., 2021) improved MIM by only taking the visible tokens as input and computing loss at the pixel level – the former change largely accelerated the training procedure as encoder's computational costs went down. The follow-up works explored different recovery targets (Wei et al., 2021), complicated model designs (Fang et al., 2022; Chen et al., 2020), and other pretext tasks.

**Our insights.** It is worth noting that MIM matches plain vision transformers very well because each token is an individual unit and only the unmasked tokens are necessary during the pre-training process. The properties does not hold for hierarchical vision transformers, making them difficult to inherit the good properties (*e.g.*, training efficiency). Although SimMIM (Xie et al., 2021b) tried to combine Swin Transformer with MIM, it uses all tokens, including those corresponding to the masked patches, shall be preserved during the encoder stage, incurring much heavier computational costs. In this paper, we design a hierarchical vision transformer that (i) integrates the advantages of ViT and Swin series and (ii) better fits MIM in terms of accuracy and speed.

## 3 HIVIT: SIMPLE AND EFFICIENT HIERARCHICAL VISION TRANSFORMER

### 3.1 PRELIMINARIES: VISION TRANSFORMERS AND MASKED IMAGE MODELING

We start with the vanilla plain vision transformer, abbreviated as ViT (Dosovitskiy et al., 2021). Mathematically, let the target model be $f(\mathbf{x}; \boldsymbol{\theta})$ where $\boldsymbol{\theta}$ denotes the learnable parameters. A training image $\mathbf{x}$ is first partitioned to a few patches, $\mathcal{X} = \{\mathbf{x}_1, \mathbf{x}_2, \ldots, \mathbf{x}_M\}$, where $M$ is the number of patches. In ViT, each image patch is transferred into a token (*i.e.*, a feature vector), and the tokens are propagated through a few transformer blocks for visual feature extraction. Let there be $L$ blocks, where the $l$-th block takes the token set of $\mathcal{U}^{(l-1)}$ as input and outputs $\mathcal{U}^{(l)}$, and

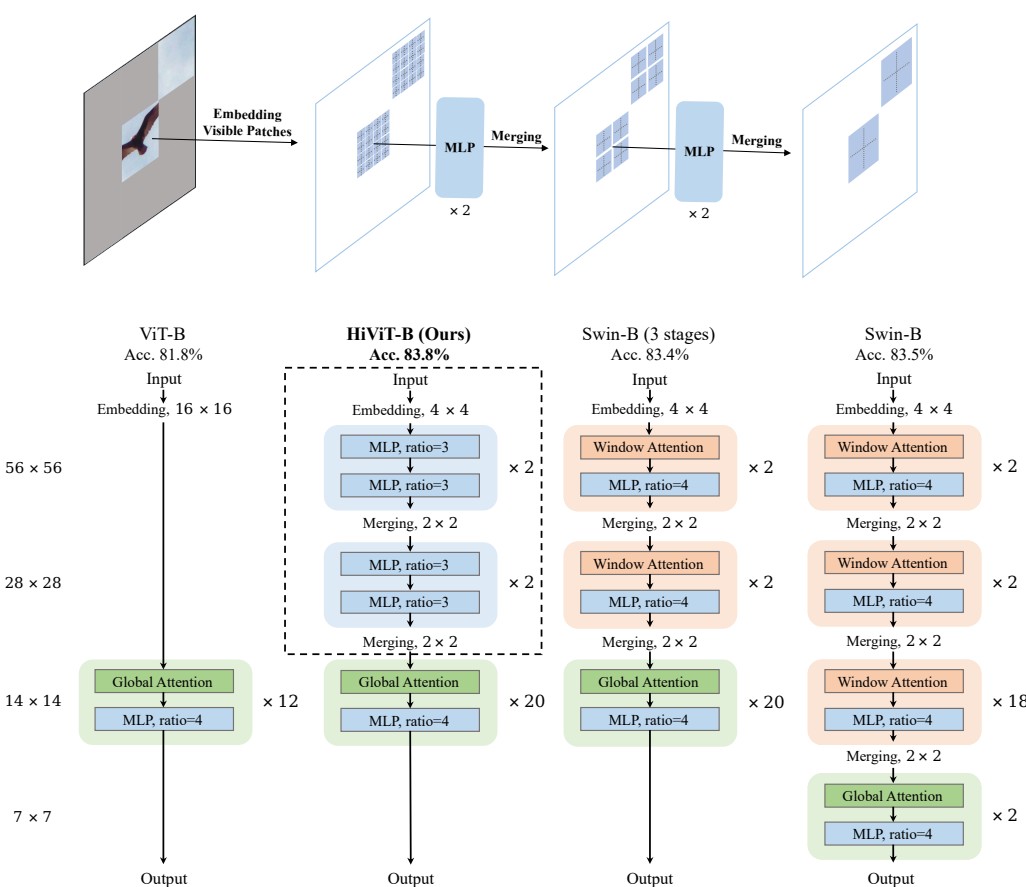

Figure 1: **Top**: hierarchical patch embedding, the way that HiViT uses for low-level feature manipulation (see the dashed box in the HiViT-B architecture). Note that the first step (*i.e.*, patch embedding) involves embedding small ($4 \times 4$) patches unlike ViT that directly embeds $16 \times 16$ patches and thus loses rich spatial information. **Bottom**: the transition from ViT to Swin, with HiViT appearing in the midst. Please refer to Table 2 for details and results.

$\mathcal{U}^{(0)} \equiv \mathcal{X}$. The main part of each block is self-attention, for which three intermediate features are computed upon $\mathbf{u}_m^{(l-1)}$, namely, the query, key, and value, denoted as $\mathbf{q}_m^{(l-1)}$, $\mathbf{k}_m^{(l-1)}$, and $\mathbf{v}_m^{(l-1)}$, respectively. Based on these quantities, the self-attention of $\mathbf{z}_m^{(l-1)}$ is computed by $\mathrm{SA}\left(\mathbf{z}_m^{(l-1)}\right) =$ $\mathrm{softmax}\left[\mathbf{q}_m^{(l-1)} \cdot \mathbf{k}_1^{(l-1)\top}, \dots, \mathbf{q}_m^{(l-1)} \cdot \mathbf{k}_M^{(l-1)\top}\right] / \sqrt{D_\mathrm{key}} \cdot \left[\mathbf{v}_1^{(l-1)}, \dots, \mathbf{v}_M^{(l-1)}\right]^\top$, where $1/\sqrt{D_\mathrm{key}}$ is a scaling vector. Auxiliary operations, including layer normalization, multi-layer perceptron, skip-layer connection, are applied after the self-attention computation.

ViT is simple yet effective on vision tasks, but lacks the ability of extracting hierarchical visual features as most CNNs (Simonyan & Zisserman, 2015; He et al., 2016; Tan & Le, 2019) have. Swin Transformer (Swin for short) is a popular variant over ViT that allows the spatial resolution to reduce gradually throughout the backbone, and avoids high computational costs with shifted-window attentions. Since then, there has been a competition between ViT and Swin – Swin showed initial advantages on detection and segmentation, yet ViT-based models managed to catch up with better implementation details (Li et al., 2022a).

The emerge of masked image modeling (MIM) becomes a new and important decider for the competition. MIM is a new paradigm of self-supervised visual representation learning that involves feeding a partially masked image to the target model $f(\mathbf{x}; \boldsymbol{\theta})$ and training the model to recover it. Following the above notations, MIM randomly chooses a subset $\mathcal{M}' \subset \{1, 2, \dots, M\}$, feeds the

patches with IDs in $\mathcal{M}'$ (denoted as $\mathcal{X}'$) into $f(\mathbf{x}; \boldsymbol{\theta})$ (*a.k.a.*, the encoder), and appends an auxiliary decoder to it, aiming at recovering the masked image contents. The objective is either computed on tokenized features (Bao et al., 2021) or pixels (He et al., 2021). In the context of MIM, ViT enjoys an efficient implementation that the tokens not in $\mathcal{X}'$ can be discarded at the beginning of encoder, while Swin must feed the entire input image, with the masked patches filled with dummy contents, into the encoder (Xie et al., 2021b). In comparison, Swin not only requires heavier computation of encoder, but also contaminates the original image data distribution with the dummy tokens.

## 3.2 Transition from ViT to Swin: Seeking for Really Useful Operations

We look forward to a variant of vision transformers that keeps the ability of learning hierarchical visual representations yet gets rid of the burdens in manipulating the MIM task. For this purpose, we inherit the methodology used in Chen et al. (2021d) to gradually transit a ViT-B model into a Swin-B model. We allow only one modification at each step so that we can distinguish useful operations. The intermediate models are illustrated in Figure 1.

A hierarchical transformer calls for an efficient function to deal with high-resolution feature maps. We investigate two possibilities here. One is the solution offered by Swin that computes shifted-window attentions – note that these operations actually banned the efficient implementation of MIM as the masked patches can no longer be discarded from input. Our solution termed hierarchical patch embedding, Figure 1 (upper), solely involves MLP and patch merging operations while does not involve either window attention or overlapped convolution. In patch merging operations, we combine $2\times2$ features along the channel dimension and introduce a full-connect layer (with learnable parameters) to convert them to a proper number of channels. As shown below, our solution achieves higher accuracy yet requires lower computational costs. More importantly, it avoids information exchange across Stage 3 tokens, which brings the efficient implementation of MIM back if the masking operation is performed on Stage 3.

Table 2: ImageNet-1K classification and ADE20K segmentation results of the models along the transition process, from ViT-B to Swin-B. "RPE" indicates the relative position embedding, "Low Att." indicates whether Window Attention is used in Stage 1 and Stage 2, "Mid Att." indicates whether Window Attention is used in Stage 3, and "T.P." is short for throughput.

| | Model | $56^2$ | $28^2$ | $14^2$ | $7^2$ | Dims | RPE | Low Att. | Mid Att. | Params | FLOPs | T.P. | IN1K Acc. | ADE mIoU |
|---|---|---|---|---|---|---|---|---|---|---|---|---|---|---|
| (o) | ViT-B | - | - | 12 | - | 768 | ✗ | ✗ | ✗ | 86.6M | 17.5G | 292 | 81.8 | 45.5 |
| (a) | (o) + Depth | - | - | 24 | - | 512 | ✗ | ✗ | ✗ | 76.7M | 15.8G | 317 | 82.9 | 48.1 |
| (b) | (a) + RPE | - | - | 24 | - | 512 | ✓ | ✗ | ✗ | 76.8M | 15.8G | 310 | 83.2 | 48.6 |
| (c) | (b) + Hierarchical | 2 | 2 | 20 | - | 512 | ✓ | ✗ | ✗ | 66.4M | 15.9G | 286 | **83.8** | **49.5** |
| (c') | (b) + Low Att. | 2 | 2 | 20 | - | 512 | ✓ | ✓ | ✗ | 66.3M | 16.0G | 275 | 83.4 | 47.9 |
| (d) | (c') + Stage 4 | 2 | 2 | 18 | 2 | 512 | ✓ | ✓ | ✓ | 87.8M | 15.5G | 278 | 83.5 | 48.0 |

We evaluate the models on ImageNet-1K classification and ADE20K segmentation, where the detailed settings are elaborated in Sections 4.1 and 4.3, respectively. For these models, the classification accuracy, the number of parameters and FLOPs are listed in Table 2. We summarize the impact of each step as follows, where $(\%, \%)$ stand for the accuracy gains compared to the prior step on ImageNet-1K and ADE20K, respectively.

**Step (a)** $(+1.1\%, +2.6\%)$ by using a deeper architecture with fewer channels. This factor contributes the largest advantage of Swin-B.

**Step (b)** $(+0.3\%, +0.5\%)$ by replacing vanilla positional encoding with relative positional encoding (RPE). This small but consistent gain validates the effectiveness of RPE.

**Step (c)** $(+0.6\%, +0.9\%)$ by introducing hierarchical patch embedding to replace the first $4$ blocks. The considerable gains validate the effectiveness of hierarchical patch embedding.

**Step (c')** $(+0.2\%, -0.7\%)$ by introducing high-resolution ($56\times56$ and $28\times28$) token maps and adding $7\times7$ shifted-window attentions. As an alternative solution (used by Swin) to Step (c), it causes a significant ADE20K segmentation accuracy and Compared to Step (c), we conclude that hierarchical patch embedding is more effective and efficient than low-level transformer blocks (with shifted-window attentions).

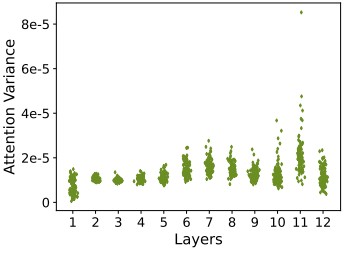 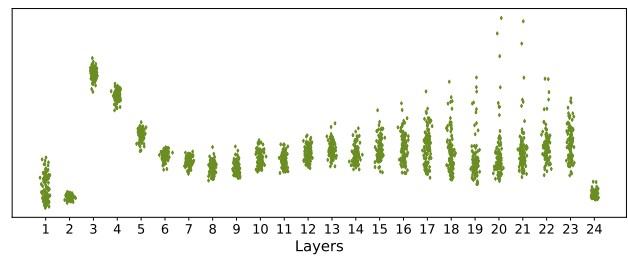

| (a) Vanilla ViT with 12 layers | (b) Deepened ViT with 24 layers |

Figure 2: As a measure of instability of self-attention, we compute the variance of attention between models trained from scratch, starting from $N = 3$ different random seeds.

Step (d) $(+0.1\%, +0.1\%)$ by adding Stage 4 (2 blocks with $7\times7$ token maps), which involves (i) reducing the number of blocks at Stage 3 by 2 (moving them to Stage 4) and (ii) modifying Stage 3 from global attentions to shifted-window attentions. This operation increases the parameter count by $33\%$ because Stage 4 contains $8\times$-dim channels while Stage 3 only has $4\times$-dim channels. Expect for the little gain, this operation this not friendly to MIM since we take Stage 3 (with $14 \times 14$ tokens) as the playground of masking patches.

From the above, we learn the lesson that, besides RPE, Swin mainly benefits from using a deeper architecture and introducing hierarchical patch embedding. Other operations, including window attentions and the Stage 4 with $7 \times 7$ tokens, do not help visual recognition. Before entering the next part, we would like to offer some side notes. If Step (a) is not performed before Step (b), *i.e.*, the first 4 layers of a 12-layer ViT-B (not a 24-layer deepened one) is replaced by hierarchical patch embedding, the performance drops. **This fact shows that early attentions can be replaced by hierarchical patch embedding only when the model is deep enough.**

To diagnose the above statement, we design an experiment on two ViTs. The first is a vanilla ViT with 12 layers and a channel number of 768, and the second is a deepened ViT with 24 layers and 512 channels. For each architecture, we train $N$ models individually from scratch (fully-supervised on ImageNet-1K). After that, we randomly sample 1,000 images from the ImageNet-1K test set, and compute the intermediate output at each transformer block – for each image at each block, it contains 12 (the head number) attention weight maps sized $196 \times 196$ ($196 = 14 \times 14$ is the token number). We first average the 12 maps into one (for each individual model), and then compute the variance of $N$ weights at each position, and finally average the $196 \times 196$ variances into a single number, indicating whether the attentions computed at this image and this block are stable – in other words, both the vanilla and deepened ViT have 1,000 variances (for 1,000 images) at each block.

In Figure 2, one can see that, in the vanilla ViT, for most samples, the variance of the average attention map at each block is small, indicating that the models have learned stable attention patterns. Some outliers show large variance in deep layers, which imply that inter-patch information is no longer needed in deep layers. In the deeper ViT, for almost all samples, attention maps of the shallow layers have much higher variance[2], indicating that these layers do not learn reliable attention maps. These observations show that early layers cannot learn effective attentions in a deeper architecture, hence removing the shallow window attentions in HiViT has little negative impacts.

### 3.3 HIVIT AND ITS APPLICATION ON MIM

Summarizing the above analysis, we obtain HiViT, the halfway architecture when Steps (a)–(c) are performed while others are discarded. An illustration of HiViT-B is shown in Figure 1. In fully-supervised ImageNet-1K classification, it achieves a higher accuracy than other variants using fewer learnable parameters.

---

[2]Exceptions happen in the first two layers, where the $16 \times 16$ patch embedding just brings in initial patch features, and we find that the attentions are less meaningful though the variances are small.

Table 3: Fully-supervised results on ImageNet-1K. The throughput of models are reported on a single Tesla-V100 GPU. † indicates a stronger baseline of ViT proposed in He et al. (2021).

| Model Type | Model | image size | Params | FLOPs | Throughput (image / s) | ImageNet-1K Top-1 Acc. |
|---|---|---|---|---|---|---|
| ConvNets | EffNet-B3 | 300 | 12M | 1.8G | 732 | 81.6 |
| | EffNet-B4 | 380 | 19M | 4.2G | 349 | 82.9 |
| | EffNet-B5 | 456 | 30M | 9.9G | 169 | 83.6 |
| | EffNet-B6 | 528 | 43M | 19.0G | 97 | 84.0 |
| | EffNet-B7 | 600 | 66M | 37.0G | 55 | 84.3 |
| Plain Transformers | DeiT-S | 224 | 22M | 4.5G | 940 | 79.8 |
| | DeiT-B | 224 | 86M | 17.5G | 292 | 81.8 |
| | ViT-B | 384 | 86M | 55.4G | 86 | 77.9 |
| | ViT-B† | 224 | 86M | 17.5G | 292 | 82.3 |
| Transformers w/ Convolution | CoAtNet-0 | 224 | 25M | 4.2G | - | 81.6 |
| | CoAtNet-1 | 224 | 42M | 8.4G | - | 83.3 |
| | CoAtNet-2 | 224 | 75M | 15.7G | - | 84.1 |
| | MViTv2-T | 224 | 24M | 4.7G | - | 82.3 |
| | MViTv2-S | 224 | 35M | 7.0G | - | 83.6 |
| | MViTv2-B | 224 | 52M | 10.2G | - | 84.4 |
| | CSWin-T | 224 | 23M | 4.3G | 701 | 82.7 |
| | CSWin-S | 224 | 35M | 6.9G | 437 | 83.6 |
| | CSWin-B | 224 | 78M | 15.0G | 250 | 84.2 |
| Hierarchical Transformers | Swin-T | 224 | 28M | 4.5G | 755 | 81.2 |
| | Swin-S | 224 | 50M | 8.7G | 437 | 83.1 |
| | Swin-B | 224 | 88M | 15.4G | 278 | 83.5 |
| | HiViT-T (ours) | 224 | 19M | 4.6G | 850 | 82.1 |
| | HiViT-S (ours) | 224 | 38M | 9.1G | 436 | 83.5 |
| | HiViT-B (ours) | 224 | 66M | 15.9G | 286 | 83.8 |

More importantly, we find that HiViT fits MIM very well. To reveal this, we show an example by following the convention to partition the entire image into $14 \times 14$ patches – each patch corresponds to a token at Stage 3. Since (i) Stages 1 and 2 are modified so that they do not contain inter-token operations and (ii) Stage 4 is removed so that all the tokens in Stage 3 are symmetric, we can use a simple implementation of MIM where all masked patches are directly discarded from input. As we shall see in experiments, this implementation not only accelerates the self-supervised learning process by nearly 100%, but also brings higher accuracy in various downstream tasks. We conjecture that part of the improvement comes from that we have cleaned input data to avoid dummy patches so that the gap between upstream and downstream image data becomes smaller.

## 4 EXPERIMENTS

We build our base model, called HiViT-B, to have a computation complexity similar to Swin-B but slightly smaller than ViT-B. We also introduce HiViT-T, HiViT-S and HiViT-L, which are versions of about $0.25\times$, $0.5\times$ and $2\times$ the computational complexity of HiViT-B, respectively. We first conduct fully supervised experiments with labels using the proposed HiViT on ImageNet-1K dataset (Deng et al., 2009). Then, HiViT models are tested using masked image modeling self-supervised methods (MIM) (He et al., 2021). The pre-trained models are also transferred to downstream tasks including object detection on COCO (Lin et al., 2014b) and semantic segmentation on ADE20K (Zhou et al., 2017). The detailed experimental settings are elaborated in Appendix A.

### 4.1 FULLY-SUPERVISED IMAGE CLASSIFICATION

**ImageNet Results.** The fully supervised training results are shown in Table 3. Compared to the vanilla ViT models, the HiViT variants report much better results in terms of classification accuracy

and computational complexity. In particular, HiViT-T/B surpass DeiT-S/B (ViT equipped with a better training strategy) by 2.3% and 2.0%, respectively, with similar computational costs. Compared to the hierarchical baseline (*i.e.*, Swin), HiViT still shows competitive performance, with HiViT-T/S/B beating Swin-T/S/B by 0.9%, 0.4%, and 0.3%, respectively, with similar computational costs. In addition, by removing shallow attentions and Stage 4 (with a large channel number), HiViT-T/S/B requires 32.2%, 24.4% and 24.4% fewer parameters compared to Swin-T/S/B models. Hence, HiViT is a simple architecture that does not require further optimization tricks beyond Swin. We will see its generalization ability in later experiments.

We note that some recent variants of vision transformers (*e.g.*, CSWin (Dong et al., 2021a) and MViTv2 (Li et al., 2022b)) achieved higher accuracy on ImageNet-1K. It should be noted that these models involved convolution operations throughout the network. The convolution operations hinder serializing the input, making them difficult to be applied to masked image modeling. We classify these models into the category of mixed models (transformers w/ convolution) in Table 3. Although HiViT is inferior to these architectures in fully-supervised ImageNet classification, we will see later that HiViT better benefits from pre-training. For example, CSWin-B (Dong et al., 2021a), with a comparable complexity to HiViT-B, reports a 84.2% top-1 accuracy on ImageNet-1K under fully-supervised learning, higher than 83.8% reported by HiViT-B, and a 50.8% AP$^{\text{box}}$ on COCO object detection. However, after pre-training using MIM and fine-tuning, HiViT-B improves these numbers to 84.6% and 53.3% (Sec. 4.3), showing its advantages in self-supervised learning.

Table 4: The fine-tuning accuracy on ImageNet-1K when the model is pre-trained without labels.

| Method | Network | Params | FLOPs | Epochs | Top-1 Acc. |
|---|---|---|---|---|---|
| BEiT (Bao et al., 2021) | ViT-B | 86M | 17.4G | 400 | 83.2 |
| CAE (Chen et al., 2022) | ViT-B | 86M | 17.4G | 800 | 83.6 |
| MaskFeat (Wei et al., 2021) | ViT-B | 86M | 17.4G | 800 | 84.0 |
| SimMIM (Xie et al., 2021b) | ViT-B | 86M | 17.4G | 800 | 83.8 |
| SimMIM (Xie et al., 2021b) | Swin-B | 88M | 15.4G | 800 | 84.0 |
| MixMIM (Liu et al., 2022) | Swin-B | 88M | 15.4G | 600 | 84.4 |
| MAE (He et al., 2021) | ViT-B | 86M | 17.4G | 1600 | 83.6 |
| Ours | HiViT-B | 66M | 15.9G | 1600 | **84.6** |
| SimMIM (Xie et al., 2021b) | Swin-L | 197M | 35.8G | 800 | 85.4 |
| SimMIM (Xie et al., 2021b) | SwinV2-H | 658M | 118.1G | 800 | 85.7 |
| MixMIM (Liu et al., 2022) | Swin-L | 197M | 35.8G | 600 | 85.7 |
| MAE (He et al., 2021) | ViT-L | 304M | 61.6G | 1600 | 85.9 |
| Ours | HiViT-L | 267M | 63.8G | 1600 | **86.4** |

## 4.2 SELF-SUPERVISED VISUAL REPRESENTATION LEARNING

**Fine-tuning.** The fine-tuning results are provided in Table 4 and only the encoder part is used to test. As shown in Table 4, under the MAE pre-training pipeline, HiViT-B achieves 1.0% performance gain over ViT-B, validating the structural advantage of our hierarchical transformer over plain vision transformers. Our results are still competitive when compared to hierarchical transformer Swin-B pre-trained using SimMIM (Xie et al., 2021b) and MixMIM (Liu et al., 2022). When scaling to the large scale, HiViT-L maintains the advantage over the plain transformer (ViT-L) with similar computational costs and fewer parameters, and claims more significant advantage (86.4% *vs.* 85.7%) compared to Swin-L (using MixMIM) and SwinV2-H (using SimMIM).

**Training Efficiency.** HiViT only requires the active (unmasked) tokens as input so that it enjoys high efficiency during the MIM pre-training. Running on eight NVIDIA Tesla V-100 GPUs, each pre-training epoch on ImageNet-1K takes 10.3 minutes, which is almost 2× as fast as the version that we feed all patches into the encoder (the case if we use a Swin-like backbone). We conjecture that this is part of the reasons why SimMIM (using Swin) reduced the input resolution to $192 \times 192$, which may result in performance drop in downstream visual recognition tasks.

## 4.3 TRANSFERRING SELF-SUPERVISED MODELS TO DENSE PREDICTION

**Objection Detection on COCO.** For objection detection, we build a Mask RCNN detector (He et al., 2017) beyond the pre-trained models. our hierarchical model can provide feature maps at

Table 5: COCO detection fine-tuning results transferred from self-supervised pre-training.

| Method | Network | Detector | Epochs | COCO AP$^{box}$ | COCO AP$^{mask}$ |
|---|---|---|---|---|---|
| MoCo v3 (Chen et al., 2021b) | ViT-B | Mask RCNN | 12 | 45.5 | 40.5 |
| BEiT (Bao et al., 2021) | ViT-B | Mask RCNN | 12 | 42.1 | 37.8 |
| PeCo (Dong et al., 2021b) | ViT-B | Mask RCNN | 12 | 44.9 | 40.4 |
| MAE (He et al., 2021) | ViT-B | Mask RCNN | 12 | 48.4 | 42.6 |
| CAE (Chen et al., 2022) | ViT-B | Mask RCNN | 12 | 49.2 | 43.3 |
| Ours | HiViT-B | Mask RCNN | 12 | **51.3** | **44.6** |
| Supervised (Li et al., 2021b) | ViT-B | Mask RCNN | 100 | 47.9 | 42.9 |
| Supervised (Dong et al., 2021a) | CSWin-B | Mask RCNN | 36 | 50.8 | 44.9 |
| Supervised (Li et al., 2022b) | MViTv2-B | Mask RCNN | 36 | 51.0 | 45.7 |
| MAE (Li et al., 2022a) | ViT-B | ViTDet | 100 | 51.2 | 45.5 |
| SimMIM (Xie et al., 2021b) | Swin-B | Mask RCNN | 36 | 52.3 | - |
| MixMIM (Liu et al., 2022) | MixMIM-B | Mask RCNN | 36 | 52.2 | 46.5 |
| Ours | HiViT-B | Mask RCNN | 36 | **53.3** | **47.0** |

multiple resolutions as inputs for the FPN head, leading to a significant advantage of HiViT over the plain transformers. As shown in Tab. 5, we compare the performance with state-of-the-art methods. When fine-tuned for 12 epochs ($1\times$), HiViT-B significantly outperforms ViT-B (pre-trained by different methods), reached an AP$^{box}$ of 51.3% and an AP$^{mask}$ of 44.6%. When fine-tuned for 36 epochs ($3\times$), HiViT-B still outperforms Swin-B pre-trained by other methods by a $\sim$1.0% AP.

**Semantic Segmentation on ADE20K.** The results on ADE20K are shown in Table 6. We report the mean intersection over union (mIoU) values. Similar to the situation on COCO, HiViT-B benefits from hierarchical features and reports a 52.8% mIoU, surpassing ViT-B pre-trained by MoCo-v3 (Chen et al., 2021c), BEiT (Bao et al., 2021), CAE (Chen et al., 2022), and MAE (He et al., 2021) by a significant margin of at least 4.0%. In terms of hierarchical vision transformers, HiViT-B outperforms MixMIM-B and reports the same performance as SimMIM-B. On these dense prediction benchmarks, the advantage of HiViT (over ViT) becomes much more significant, validating the usefulness of hierarchical visual representations.

Table 6: Semantic segmentation results on ADE20K.

| Method | Network | mIoU |
|---|---|---|
| Supervised (He et al., 2021) | ViT-B | 47.0 |
| MoCo v3 (Chen et al., 2021b) | ViT-B | 47.3 |
| BEiT (Bao et al., 2021) | ViT-B | 47.1 |
| PeCo (Dong et al., 2021b) | ViT-B | 48.5 |
| MAE (He et al., 2021) | ViT-B | 48.1 |
| CAE (Chen et al., 2022) | ViT-B | 48.8 |
| SimMIM (Xie et al., 2021b) | Swin-B | 52.8 |
| MixMIM (Liu et al., 2022) | MixMIM-B | 50.3 |
| Ours | HiViT-B | **52.8** |

## 5 CONCLUSIONS

This paper presents a hierarchical vision transformer named HiViT. Starting with Swin Transformers, we remove redundant operations that cross the border of tokens in the main stage, and show that such modifications do not harm, but slightly improve the model's performance in both fully-supervised and self-supervised visual representation learning. HiViT shows a clear advantage in integrating with masked image modeling, on which the efficient implementation of MAE can be directly transplanted from ViT to HiViT, accelerating the training speed significantly. We expect that HiViT becomes an off-the-shelf replacement of ViT and Swin in the future research, especially when the models need to be pre-trained with masked image modeling.

**Limitations.** Despite the improvement observed in the experiments, our method has some limitations. The most important one lies in that the masking unit size is fixed – this implies that we need to choose a single 'main stage'. Fortunately, the 3rd stage of Swin Transformers contribute most parameters and computations, hence it is naturally chosen, however, the method may encounter difficulties in the scenarios that no dominant stages exist. In addition, we look forward to advanced architecture designs that go beyond the constraints. A possible solution lies in modifying low-level code (*e.g.*, in CUDA) to support arbitrary and variable grouping of tokens, but, more essentially, we expect a flexible learning framework beyond MIM that supports variable sizes of masking units.

## ETHICS STATEMENT

Our research focuses on two fundamental topics in computer vision, namely, (i) designing an efficient architecture of deep neural networks and (ii) self-supervised visual representation learning. The datasets used for evaluation are public benchmarks that were widely used in the community. To the best of our knowledge, this research does not have ethical concerns to report.

## REPRODUCIBILITY STATEMENT

We have included the code in the supplementary material. After the review process, we will also open-source the pre-trained models (which can also be obtained by running our code on public datasets, *e.g.*, ImageNet-1K) to ease the community to reproduce our results.

## ACKNOWLEDGEMENT

This work was supported by National Natural Science Foundation of China (NSFC) under Grant 62225208, 62171431 and 61836012.

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

## A   APPENDIX

Table 7: Configurations for HiViT variants.

| Model | Depth $56^2$ | $28^2$ | $14^2$ | Dim $56^2$ | $28^2$ | $14^2$ | Heads $56^2$ | $28^2$ | $14^2$ | Params (M) | FLOPs (G) |
|---|---|---|---|---|---|---|---|---|---|---|---|
| HiViT-T (tiny) | 1 | 1 | 10 | 96 | 192 | 384 | - | - | 6 | 19.2 | 4.6 |
| HiViT-S (small) | 2 | 2 | 20 | 96 | 192 | 384 | - | - | 6 | 37.5 | 9.1 |
| HiViT-B (base) | 2 | 2 | 20 | 128 | 256 | 512 | - | - | 8 | 66.4 | 15.9 |
| HiViT-L (large) | 4 | 4 | 36 | 192 | 384 | 768 | - | - | 12 | 266.8 | 63.8 |

**Model Configurations**   Three models HiViT-T/S/B/L are trained through supervised learning or self-supervised learning with configurations included in Tab. 7. The "Depth" represents the block number on different stages ($56^2$, $28^2$, and $14^2$ are the 1st, 2nd, and 3rd stages respectively). "Dim" and "Heads" respectively denote the dimension and attention head numbers. We align the models according to FLOPs while using fewer parameters than the compared methods. We report the inference throughput speed in Tab. 3 by testing the $224^2$ images using a V100 GPU.

**Experimental Details for Supervised Learning**   We first evaluate HiViT with supervised learning on ImageNet-1K (Deng et al., 2009) which contains 1.28M training images and 50K validation ones divided into 1,000 categories. We follow Swin (Liu et al., 2021) and use the same training settings. Specifically, we use the AdamW optimizer (Loshchilov & Hutter, 2017) with an initial learning rate 0.001, a weight decay 0.05, batch size 1024, a cosine decay learning rate scheduler, and a linearly warm-up for 20 epochs. All the models are trained for 300 epochs with augmentation and regularization strategies (Liu et al., 2021) and exponential moving average (EMA). The input size is $224 \times 224$ by default. The output feature of 3rd stage is followed by an average pooling layer and then a classifier layer. The drop path rates 0.05, 0.3, and 0.5 are respectively used for HiViT-T/S/B.

**Experimental Details for Self-supervised Learning**   For self-supervised pre-training, the ImageNet-1K training dataset without labels is used. The pre-trained models is fine-tuned and tested on the validation dataset. The pre-training settings follow those of MAE (He et al., 2021). Specifically, the mask ratio is set to 75% in default. The normalized target trick is also adopted. In base model, we use the AdamW optimizer (Loshchilov & Hutter, 2017) with an initial learning rate of $1.0 \times 10^{-4}$, a weight decay of 0.05, and a learning rate follows the cosine decay learning schedule with a warm-up for 40 epochs. The batch size is set to 4096 and the input size is $224 \times 224$. The overall pipeline is an encoder-decoder framework where the decoder is designed to have 6 transformer layers followed by a reshape operation to cast the feature to $3 \times 224 \times 224$. Random cropping and random horizontal flip are used for data augmentation. When fine-tuning, we follow the settings from  (He et al., 2021) where the models are trained for 100 epochs using the AdamW optimizer with a warm-up for 5 epochs, a weight decay 0.05, and the input size $224 \times 224$. We use the layer-wise learning rate decay 0.85. The initial learning rate is set to $5 \times 10^{-4}$ and batch size is set to 1024.

**Experimental Details for Down-stream Tasks**   We transfer the self-supervised pre-trained models to object detection on MS COCO and semantic segmentation on ADE20K. We use the Mask R-CNN (He et al., 2017) head implemented by the MMDetection library (Chen et al., 2019). On the MS COCO dataset, we adopt the AdamW optimizer (Loshchilov & Hutter, 2017) with an initial learning rate of $1 \times 10^{-4}$ which decays by $10 \times$ after the 9-th and 11-th epochs when $1 \times$ training schedule (12 epochs) is adopted. The layer-wise decay rate is set to 0.9 and the multi-scale training and single-scale testing strategies are used. For ADE20K, we employ the UperNet (Xiao et al., 2018) head following BEiT (Bao et al., 2021). The AdamW optimizer with a learning rate $1 \times 10^{-4}$ to train the model for 80k iterations with a batch size 32.

Table 8: Detailed comparison of the pre-training time and GPU memory cost.

| Pre-train | Model | Input Size | GPU Memory | Pre-training time (mins pre epoch) |
|---|---|---|---|---|
| MAE (He et al., 2021) | ViT-B | 224×224 | 15282M | 8.8 |
| SimMIM (Xie et al., 2021b) | Swin-B | 192×192 | 18245M | 14.3 |
| SimMIM (Xie et al., 2021b) | Swin-B | 224×224 | 25710M | 18.7 |
| Ours | HiViT-B | 224×224 | 17906M | 10.3 |

Table 9: Ablation studies on shallow stage depth.

| Module # | | | Params (M) | FLOPs (G) | ImageNet Acc. |
|---|---|---|---|---|---|
| $56^2$ | $28^2$ | $14^2$ | | | |
| 0 | 0 | 24 | 77.1 | 16.0 | 84.1 |
| 1 | 1 | 22 | 71.8 | 15.9 | 84.5 |
| 2 | 2 | 20 | 66.4 | 15.9 | 84.5 |
| 3 | 3 | 18 | 61.1 | 15.8 | 84.4 |
| 4 | 4 | 16 | 55.8 | 15.7 | 84.1 |

**Comparison of the Pre-training Time and GPU Memory Cost** We test on an $8\times$V100-SXM2 platform with PyTorch 1.8.0 and CUDA 10.2. All the methods use a batch size of 1024. As shown in Tab. 8, compared to MAE, SimMIM requires more memory and pre-training time because all the image patches (including the visible and the mask patches) need to be computed. With the same input resolution, the proposed HiViT models requires significantly shorter pre-training time per epoch and smaller GPU memory than SimMIM's.

**Depth of Shallow Stages** We ablate the effect of shallow stage depth, Tab. 9, by setting different block numbers for stage 1, 2, and 3 ($56^2$, $28^2$, and $14^2$ respectively). The best setting (2–2–20 block) achieves $84.5\%$ performance. Changing the depth of shallow stages to be smaller or larger leads to worse performance.

