# OpenReview forum: "HiViT: A Simpler and More Efficient Design of Hierarchical Vision Transformer"
_ICLR.cc/2023/Conference — ICLR 2023 notable top 25%_

### Official Review · Reviewer_vrWv · 2022-10-23

**Confidence:** 5
**Correctness:** 3
**Technical Novelty And Significance:** 3
**Empirical Novelty And Significance:** 4
**Recommendation:** 8

**Clarity, Quality, Novelty And Reproducibility:**

The paper has some minor clarity issues describing the details of the method. Overall quality is good. Novelty and contributions are enough for the community as it introduces some key findings, and the method can be widely adapted. Reproducibility is good as method is simple and code is provided.

**Strength And Weaknesses:**

Strength:
1.	The paper provides a detailed study regarding prevail Swin-Transformer architecture, identifying the key parts for its superior performance v.s. plain ViT, e.g increasing depth and decrease channels, add relative position embedding.
2.	The paper proposes an alternative way to enjoy the performance gain from hierarchical ViT which is simpler and friendly for techniques that are designed for plain ViT, i.e. Mask Image Modeling.
Weaknesses:
1.	There are several part of descriptions not clear enough.
a.	Table 2, no explanation for RPE, Low Att, Mid Att, etc in table caption.
b.	Stage definition missing, though can be referred, but better have clear definition.
c.	Can authors also add Param and Flops to Table 2, as changing components will change computation as well.
d.	Incomplete sentences: Page 5, Step (c’) ‘causes a significant accuracy’;  Sec 3.3 ‘all tokens in Stage ? are symmetric’
e.	Merging operation details? One can only infer it from figure 1, can authors add some descriptions about it?
2.	Regarding the proposed hierarchical patch embedding, can authors give more explanation about the specifical design as 2 consecutive MLP with ratio 3, how authors reach this design. As this is a critical module proposed, can authors provide some ablation studies about it, e.g. why uses a different MLP ratio of 3 as latter stages uses 4 instead? Will allocate computations differently to 56x56, 28x28 influence the final performance? What computation percentage should be used in patch embedding v.s. latter stage (14x14) for better performance?



**Summary Of The Paper:**

This paper investigates the improvement from plain ViT to hierarchical Swin-Transformer step-by-step and proposes a new hierarchical vision transformer that only uses hierarchical patch embedding at the beginning and keeps the global attention at 14x14 level instead of window attention. It shows similar or better performance than Swin transformers on Imagenet classification, segmentation and detection.

**Summary Of The Review:**


Overall, the paper presents a good study of plain ViT and hierarchical ViT, and provides a simpler hierarchical ViT with similar or better vision task performance and is friendly to many follow-up techniques of plain ViT. The proposed method can be widely adapted with its simple design and good performance. It would be much better if authors can provides further studies regarding the proposed patch embedding module.

---

> ### Author Response · Authors · 2022-11-18
> **Response to reviewer vrWv**
>
> Thanks for your comments that help us improve this work. We hope our responses can address your concerns. Further comments are welcomed.
>
> **Comment 1:** There are several parts of the descriptions not clear enough. a) Table 2, no explanation for RPE, Low Att, Mid Att, etc in table caption. b) Stage definition missing, though can be referred, better have a clear definition. c) Can authors also add Params and FLOPs to Table 2, as changing components will change computation as well. d) Incomplete sentences: Page 5, Step (c') 'causes a significant accuracy'; Sec 3.3 'all tokens in Stage ? are symmetric' e) Merging operation details? One can only infer it from figure 1, can authors add some descriptions about it?
>
> **Response 1:** Thanks for the comments that make the descriptions clearer.
>
> a) We have added the explanation for RPE, Low Att, Mid Att: "RPE indicates the relative position embedding, Low Att. indicates whether Window Attention is used in Stage 1 and Stage 2, Mid Att. indicates whether Window Attention is used in Stage 3, and T.P. is short for throughput".
>
> b) We have added a footnote: "Stage refers to components that process the same resolution in hierarchical models. In this paper, Stage 1, Stage 2, Stage 3, and Stage 4 respectively refer to components that process the resolutions of $56^2$, $28^2$, $14^2$, and $7^2$ in ImageNet classification" in Page 2.
>
> c) We have added runtime (in throughput), FLOPs, and Params to the revised paper. Please kindly refer to Table 2.
>
> d) We have revised the sentence in Page 5 to "As an alternative solution (used by Swin) to Step (c), it achieves a higher segmentation accuracy on ADE20K." We have revised the sentence in Sec 3.3 into "all the tokens in Stage 3 are symmetric." We would like to perform a careful proof-reading through the paper to solve the minor issues.
>
> e) We have added merging operation details as "In merging operation, following Swin, we combine 2 $\times$ 2 features along the channel dimension and introduce a fully-connected layer (with learnable parameters) to convert them to a proper number of channels." in Sec 3.2.
>
> **Comment 2:** Regarding the proposed hierarchical patch embedding, can authors give more explanation about the specific design as 2 consecutive MLP with ratio 3, how authors reach this design. As this is a critical module proposed, can authors provide some ablation studies about it, e.g. why uses a different MLP ratio of 3 as latter stages use 4 instead? Will allocating computations differently to 56x56, 28x28 influence the final performance? What computation percentage should be used in patch embedding v.s. latter stage (14x14) for better performance?
>
> **Response 2:** We provide explanations and ablations as follows.
>
> a) We determined the MLP ratio in the consideration of computational costs, not recognition accuracy. Specifically, let $B$, $N$, and $C$ denote the batch size, number of patches, and number of channels, respectively. With an MLP ratio of $3$, the FLOPs of an FC layer is $12\times BNC^2$, which aligns with the vanilla transformer block. Hence, we did not perform ablations with an MLP ratio of $4$, which increases the FLOPs to $16\times BNC^2$.
>
> b) The percentage of $56\times56$ and $28\times28$ stages does have significant impacts on the recognition accuracy. We have added the following ablative results to the Appendix.
>
> | Module #        | Params  | FLOPs   | ImageNet Acc.  |
> | :-------------: | :-----: | :-----: | :-----------: |
> | 0 - 0 - 24          | 77.1M   | 16.0G   | 84.1%         |
> | 1 - 1 - 22          | 71.8M   | 15.9G   | 84.5%         |
> | 2 - 2 - 20          | 66.4M   | 15.9G   | 84.5%         |
> | 3 - 3 - 18          | 61.1M   | 15.8G   | 84.4%         |
> | 4 - 4 - 16          | 55.8M   | 15.7G   | 84.1%         |

---

> > ### Comment · Reviewer_vrWv · 2022-12-12
> > **Thanks for the response**
> >
> > Thank the authors for the response. I think overall the empirical results can benefit the community regarding understanding Hierarchy ViTs. I vote to accept this paper.

---

### Official Review · Reviewer_5NXC · 2022-10-24

**Confidence:** 4
**Correctness:** 4
**Technical Novelty And Significance:** 2
**Empirical Novelty And Significance:** 3
**Recommendation:** 6

**Clarity, Quality, Novelty And Reproducibility:**

The clarity is good. The illustration in Figure 1 along with ablations in Table 2 makes the main message clear. The quality is reasonable, too. The authors test on state-of-the-art settings and compare with state-of-the-art methods with extensive experiments. The novelty isn't particularly high, but can be a useful reference for future researchers.

**Strength And Weaknesses:**

Strengths:
+ Swins, ViTs, and MAE are state-of-the-art and widely-used tools in computer vision. The three main contributions, (1) understanding what makes Swin works well, (2) advancing architecture design, and (3) showing that HiViT works with masked-pretraining, are practically useful. I can see that many computer vision practitioners will be interested in this paper.
+ The ablations in Table 2 are clear and convincing.

Weaknesses:
- While HiViT works well with masked-pretraining, without masked pre-training it is not very strong; For example, MViTv2 is more accurate and uses only 2/3 of FLOPs. This makes HiVIT's usefulness in practice limited.
- Many ablations are not entirely new. For example, the effectiveness of RPE and hierarchical resolutions have been shown in the original Swin paper. The finding that I find novel is that C512 x 24L works quite a lot better than C768 x 12L. However, this is technically a small change. I thus overall would not rate the technical contributions high.
- Adding runtime, flops, and params into Table 2 would makes the trade-offs clearer.

- minor points on writing:
1) why use the word "merging" instead of "pooling", which is more concretely defined?


**Summary Of The Paper:**

In this paper, the authors study what makes hierarchical ViTs (in particular SwinTransformers) work, and based on the insights, propose a new vision transformer architecture called HiViT. The authors show that HiViT works better than Swin and vanilla ViT on ImageNet and ADE20K. Moreover, HiViT doesn't use window attention, makes it compatible to MAE style efficient SSL pre-training. The authors show that masked pre-training on HiViT obtains better accuracy than MAE on vanilla ViTs.

**Summary Of The Review:**

While the technical novelty of this paper is somewhat limited, I still think this paper can be a useful reference for practitioners and future researchers on architecture design. Overall, I rate this paper as marginally above the acceptance threshold.

---

> ### Author Response · Authors · 2022-11-18
> **Response to reviewer 5NXC**
>
> Thanks for your comments that help us improve this work. We hope our responses can address your concerns. Further comments are welcomed.
>
> **Comment 1:** While HiViT works well with masked pre-training, without masked pre-training it is not very strong. For example, MViT-v2 is more accurate and uses only 2/3 of FLOPs. This makes HiVIT's usefulness in practice limited.
>
> **Response 1:** Good question! HiViT was designed for the pipeline of MIM pre-training followed by fine-tuning. This pipeline has shown significant advantages over direct supervised learning. In the case of HiViT vs MViT-v2: (1) On ImageNet, HiViT with pre-training outperforms MViT-v2 by 0.2% (84.6% vs. 84.4%). (2) On COCO object detection, HiViT with pre-training outperforms MViT-v2 by a much larger gain of 2.3% (53.3% vs. 51.0%). We believe that further study can be made upon HiViT to reduce computational costs.
>
> **Comment 2:** Many ablations are not entirely new. For example, the effectiveness of RPE and hierarchical resolutions have been shown in the original Swin paper. The finding that I find novel is that C512 x 24L works quite a lot better than C768 x 12L. However, this is technically a small change. I thus overall would not rate the technical contributions high.
>
> **Response 2:** The main contribution of this work is to design an MIM-friendly hierarchical vision transformer. For this purpose, we absorb good practices from both ViT (being MIM-friendly) and Swin (being hierarchical). The ablations (step-by-step change from ViT to Swin) are not new, yet we achieve an efficient architecture, HiViT, in the midst of transition and validate its superiority in MIM. HiViT has the potential of being a generalized architecture for visual recognition, which supports our technical contribution.
>
> **Comment 3:** Adding runtime, FLOPs, and Params into Table 2 would make the trade-offs clearer.
>
> **Response 3:** As suggested, we have added runtime (in throughput), FLOPs, and Params to Table 2 of the revised paper.
>
> **Comment 4:** [minor] Why use the word "merging" instead of "pooling", which is more concretely defined?
>
> **Response 4:** The word "patch merging" is inherited from the paper of Swin transformer. Actually, "merging" is a preciser word than "pooling" in this paper. "pooling", such as avg-pooling or max-pooling, is conventionally defined as a down-sampling operation without using learnable parameters. In merging operation, following Swin, we combine 2 $\times$ 2 features along the channel dimension and introduce a fully-connected layer (with learnable parameters) to convert them to a proper number of channels.

---

> > ### Comment · Reviewer_5NXC · 2022-12-01
> > **Re: Response to reviewer 5NXC**
> >
> > Thanks for the response. The additional information on runtime, FLOPs, and Params are helpful. After reading other reviews, I still think despite the simplicity, the empirical results provided can be useful for our community in designing strong and flexible vision architectures. I keep my recommendation unchanged.

---

### Official Review · Reviewer_mVGH · 2022-10-25

**Confidence:** 5
**Correctness:** 3
**Technical Novelty And Significance:** 3
**Empirical Novelty And Significance:** 2
**Recommendation:** 5

**Clarity, Quality, Novelty And Reproducibility:**

The paper presents the ideas clearly and the work is original. The quality can be improved if the concerns in the weaknesses part are addressed.

**Strength And Weaknesses:**

Strength:
- The paper is well written, the contributions are clearly stated and supported by empirical experimental results and analysis.
- The paper gives a step-by-step analysis on the differences between ViT and Swin, which
- The proposed model addresses the problem of incompatibility between MAE and hierarchical vision transformers, by simply removing the 7x7 stage and the self-attention in early stages.
- The authors performed rich experiments to validate the effectiveness of the proposed HiViT. The training and design details are provided, and the overall method seems reproducible.

Weaknesses:
- This paper claims that early attentions can be replaced by hierarchical patch embedding only when the model is deep enough, with an attention variance comparison on the deepened plain ViT. However, as the proposed model removed the self-attention on hierarchical ViT (Swin) instead of plain ViT, does the statement still hold on Swin?
- In the analysis section, the authors proposed a variant of Swin without stage4 and does not hurt performance. However, the model without stage4 also used global window in stage 3 (as illustrated in Fig1, bottom, 3rd & 4th figures), which may also have a significant impact on the performance and computational cost of the model. Therefore, the experiment may not well support the claim that stage4 is not helpful.
- The overall architecture of HiViT comprises an overlap patch embedding that embeds the image to 14x14 and use continuous transformer blocks like plain ViTs. Yet, using hierarchical patch embedding have been proposed by several papers, e.g., papers use overlapping patch embed [1-4], and another similar work that also removed the self-attention in early stages, and only use MLP in 56x56 and 28x28 [4].
[1] Xiao, Tete, et al. "Early convolutions help transformers see better." Advances in Neural Information Processing Systems 34 (2021): 30392-30400.
[2] Guo, Jianyuan, et al. "Cmt: Convolutional neural networks meet vision transformers." Proceedings of the IEEE/CVF Conference on Computer Vision and Pattern Recognition. 2022.
[3] Ali, Alaaeldin, et al. "Xcit: Cross-covariance image transformers." Advances in neural information processing systems 34 (2021): 20014-20027.
[4] Pan, Zizheng, et al. "Less is more: Pay less attention in vision transformers." Proceedings of the AAAI Conference on Artificial Intelligence. Vol. 36. No. 2. 2022.
- Since an important claim of this paper is enabling hierarchical architectures to trained with MAE and save pretraining cost, a more detailed comparison of the training time or GPU usage, shall be given for a better reflection of the capacity of the architecture for pretraining, as an example, the Table2 in [5].
[5] Huang, Lang, et al. "Green Hierarchical Vision Transformer for Masked Image Modeling." arXiv preprint arXiv:2205.13515 (2022).
- The performance gain in Table 4 may not well reflect whether HiViT has “structural advantage over plain vision transformers”. ViT-B has 77.9% acc while HiViT has 83.8% acc in the supervised training, while MAE+ViT-B has 83.6%(+5.7%) acc and MAE+HiViT has 84.6%(+0.8%) in the pretraining setting. Therefore, it seems HiViT do not gain much from the pretraining, even with 1600 epochs. This phenomenon should be further explained.
- There are some other minor issues, such as mistakes in writing. For example, “As an alternative solution (used by Swin) to Step (c), it causes a significant ADE20K segmentation accuracy and Compared to Step (c)”.


**Summary Of The Paper:**

This paper proposes a hierarchical ViT, by analyzing the effective design in Swin compared with the plain ViT. The model contains a hierarchical patch embedding with only MLP used in early stages, using transformer blocks with global attention in the 14x14 stage, and removing the last stage. Experiments on ImageNet, COCO and ADE20K have shown quite promising results. The architecture is also compatible with MAE, which can achieve higher accuracy with the mask pretraining.

**Summary Of The Review:**

Overall, I think this paper is well organized. It studies an important task, where the hierarchical models are not well compatible with the powerful MAE. The analysis of the plain and hierarchical ViTs are quite impressive.

However, this proposed model is a simple plain transformer with hierarchical patch stem, which has been used in several prior works. Hence, the technical novelty is limited. The key contribution of using MAE for hierarchical model pretraining, does not give impressive results. The experiments do not present the actual training cost saved with HiViT, meanwhile the improvement with 1600epoch MAE pretraining only bring limited performance improvement compared with ViT.

Therefore, the reviewer votes for the borderline in the first round.

---

> ### Author Response · Authors · 2022-11-18
> **Response to reviewer mVGH (2/2)**
>
> **Comment 4:** Since an important claim of this paper is enabling hierarchical architectures to train with MAE and save pre-training cost, a more detailed comparison of the training time or GPU usage, shall be given for a better reflection of the capacity of the architecture for pre-training.
>
> **Response 4:** As suggested, we have added a detailed comparison of the training time and GPU memory in Appendix of the revised paper. We tested on a 8 $\times$ V100-SXM2 platform with pytorch 1.8.0 and CUDA 10.2. All methods are tested using a batch size of 1024.
>
> | Method | Model     | Input Size | Memory | Time per Epoch (mins) |
> | :---- | :-------: | :--------: | :----: | :----------: |
> | MAE    | ViT-B     | 224        | 15282M | 8.8          |
> | SimMIM | Swin-B    | 192        | 18245M | 14.3         |
> | SimMIM | Swin-B    | 224        | 25710M | 18.7         |
> | Ours   | HiViT-B   | 224        | 17906M | 10.3         |
>
> **Comment 5:** The performance gain in Table 4 may not well reflect whether HiViT has a "structural advantage over plain vision transformers". ViT-B has 77.9% acc while HiViT has 83.8% acc in the supervised training, MAE+ViT-B has 83.6% (+5.7%) acc and MAE+HiViT has 84.6% (+0.8%) in the pre-training setting. Therefore, it seems HiViT does not gain much from the pre-training, even with 1600 epochs. This phenomenon should be further explained.
>
> **Response 5:** The ViT-B with a 77.9% acc comes from an early baseline, which did not use either sophisticated data augmentation nor optimal training hyper-parameters. When using a strong baseline (as in the MAE paper), the acc of ViT-B reaches 82.3%, *i.e.*, MAE+ViT-B has +1.3% acc gain (83.6% vs. 82.3%). At a higher baseline (HiViT-B with a 83.8% acc in supervised learning), the performance gain (+0.8%) of MAE+HiViT in the pre-training setting is as expected. To clarify this point, we have included the strong baseline in Table 3. In addition, we note that such gain is achieved by using fewer parameters (66M vs. 87M) and computational costs (15.9 GFLOPs vs. 17.4 GFLOPs) than ViT-B. During rebuttal, we add an experiment to compare the gains of HiViT-B and ViT-B under similar computational costs (see the table below). With 4 more transformer blocks, the computational complexity of HiViT-B' is on par with ViT-B. Interestingly, the supervised acc does not change, but the self-supervised acc goes up by 0.4%, leading to an overall +1.2% acc gain (85.0% vs 83.8%).
>
> | Model       | Depth         | Params        | FLOPs        | Supervised acc | Self-Supervised acc  |
> | :--------- | :-----------: | :-----------: | :----------: | :------------: | :------------------: |
> | ViT-B       | 12            | 86.6M         | 17.5G        | 82.3%          | 83.6% (+1.3%)        |
> | HiViT-B     | 2 - 2 - 20        | 66.4M         | 15.9G        | 83.8%          | 84.6% (+0.8%)        |
> | HiViT-B'    | 2 - 2 - 24        | 79.1M         | 18.5G        | 83.8%          | 85.0% (+1.2%)        |
>
>
> **Comment 6:** There are some other minor issues, such as mistakes in writing. For example, "As an alternative solution (used by Swin) to Step (c), it causes a significant ADE20K segmentation accuracy and Compared to Step (c)".
>
> **Response 6:** As suggested, we have revised the sentence to "As an alternative solution (used by Swin) to Step (c), it achieves higher segmentation performance on ADE20K." We have also carefully proofread the paper to fix minor issues.

---

> ### Author Response · Authors · 2022-11-18
> **Response to reviewer mVGH (1/2)**
>
> Thanks for your comments that help us improve this work. We hope our responses can address your concerns. Further comments are welcomed.
>
> **Comment 1:** This paper claims that early attention can be replaced by hierarchical patch embedding only when the model is deep enough, with an attention variance comparison on the deepened plain ViT. However, as the proposed model removed the self-attention on hierarchical ViT (Swin) instead of plain ViT, does the statement still hold on Swin?
>
> **Response 1:** Thanks for the good question. The final architecture, HiViT, is more similar to ViT rather than Swin [Footnote a], so our observation on ViT applies well to HiViT. Indeed, the phenomenon (*i.e.*, attention variance comparison) does not show up on Swin, arguably because the window attention mechanism has changed the property. Yet, this does not impact our conclusions.
>
> [Footnote a]: HiViT adds a hierarchical patch embedding before the first transformer block of ViT. On the other hand, it removes adjusts much more designs of Swin, so we claim that HiViT is more similar to ViT.
>
> **Comment 2:** In the analysis section, the authors proposed a variant of Swin without Stage4 which does not hurt performance. However, the model without Stage4 also used a global window in Stage3 (as illustrated in Fig1, bottom, 3rd & 4th figures), which may also have a significant impact on the performance and computational cost of the model. Therefore, the experiment may not well support the claim that Stage4 is not helpful.
>
> **Response 2:** We provide more results and comparisons to show that Stage4 is not helpful. (1) We conduct an experiment by adding Stage4 to HiViT while retaining using global attentions on Stage3. The accuracy on ImageNet remains the same with that of HiViT (83.8% top-1 acc), implying that Stage4 does not improve visual recognition. (2) By removing Stage4 and using global attention in Stage3, we significantly reduced the parameter number (66M vs. 86M) at a small increase of FLOPs (16.0G vs. 15.4G). So, overall, removing Stage 4 has reduced computational complexity. In the revision, we have added parameter counts and FLOPs to Table 2.
>
> **Comment 3:** The overall architecture of HiViT comprises an overlap patch embedding that embeds the image to 14x14 and uses continuous transformer blocks like plain ViTs. Yet, using hierarchical patch embedding has been proposed by several papers [1-4].
>
> **Response 3:** Thanks for the reminder. We have cited these papers in the final version with necessary discussions.
>
> (1) The core purpose of this work is to design a hierarchical vision transformer that applies to MIM. The conventional hierarchical patch embedding methods ([1-4], implemented using window attentions or overlapped convolution), albeit being effective for visual recognition, suffer from information leak and thus are difficult to be integrated with masked image modeling (MIM). Specifically, [1] added convolution to the patch embedding and [2,3] added convolution to each transformer block, introducing information leak to MIM pre-training. [4] used pure MLPs for local feature extraction, but it contains deformable convolutions that still introduced information leak.
>
> (2) HiViT, for the first time, solves this conflict and leads to a very simple implementation of hierarchical patch embedding. Specifically, HiViT does **not** use window attentions or overlapped convolutions. To further clarify the difference, we have added a sentence "our solution, termed hierarchical patch embedding, solely involves MLP and patch merging operations and does not involve either window attentions or overlapped convolutions to avoid information leak." to Section 3.2.
>
> [1] Xiao, Tete, et al. "Early convolutions help transformers see better." Advances in Neural Information Processing Systems 34 (2021): 30392-30400.
>
> [2] Guo, Jianyuan, et al. "Cmt: Convolutional neural networks meet vision transformers." Proceedings of the IEEE/CVF Conference on Computer Vision and Pattern Recognition. 2022.
>
> [3] Ali, Alaaeldin, et al. "Xcit: Cross-covariance image transformers." Advances in neural information processing systems 34 (2021): 20014-20027.
>
> [4] Pan, Zizheng, et al. "Less is more: Pay less attention in vision transformers." Proceedings of the AAAI Conference on Artificial Intelligence. Vol. 36. No. 2. 2022.

---

> ### Author Response · Authors · 2022-12-10
> **Looking forward to post-rebuttal discussions**
>
> Dear Reviewer mVGH,
>
>
> We would like to express our sincere gratitude for your time and efforts in reviewing our paper. Your comments and feedback will be invaluable in helping us improve the final version of our paper.
>
>
> As the deadline for discussion is approaching, we are willing to provide any additional clarifications that you may require. In our previous response, we have carefully considered your comments and made detailed responses, which are summarized below:
>
>
> - We highlighted the differences between several hierarchical patch embedding methods you mentioned and HiViT.
>
> - We provide more results and comparisons to show that Stage4 is not helpful.
>
> - We explained why the performance gain in Table 4 reflect that HiViT has a "structural advantage over plain vision transformers".
>
> - We corrected the typo that you pointed out.
>
>
> We hope the above experiments and analyses could clarify your concerns. Thanks for your time and efforts!
>
>
> Best,
>
>
> Authors.

---

### Author Response · Authors · 2022-11-18
**Response to all reviewers**

We deeply appreciate the reviewers for their insightful and constructive comments. In the responses, we refer to Reviewer mVGH, Reviewer 5NXC, and Reviewer vrWv as R1, R2, and R3, respectively.

All reviewers (R1, R2, R3) acknowledged that the proposed architecture (HiViT) addressed an important problem of incompatibility between hierarchical vision transformers and masked image modeling (MIM) (*e.g.*, MAE). R1 and R2 approved that HiViT was tested on standard visual recognition benchmarks and compared with state-of-the-art methods with extensive experiments.

Also, the reviewers pointed out that "many computer vision practitioners will be interested in this paper" (R2) and "the proposed method can be widely adapted" (R3), as the ablations on hierarchical architecture are "clear and convincing" (R2).

The main concerns involve two parts. First, the comparison against prior methods. R1 raised a few related work: we thank him/her for the reminder, and point out that HiViT is different from them and enjoys clear advantages in the application to MIM. R2 raised MViT-v2 as another efficient architecture: we point out that MViT-v2 is not easily integrated with MIM and hence the downstream recognition accuracy is inferior to HiViT. Second, some technical details or choices to be further elaborated, where we respond as follows.

Again, thank you very much for approving the contributions and potential of this study. In what follows, we respond to the reviewers' comments one by one. We have also attached a revised manuscript with the promised contents added.

---

### Decision · Program_Chairs · 2023-01-20

**Decision:**

Accept: notable-top-25%

**Justification For Why Not Higher Score:**

This paper is a strong empirical paper; however, it does not necessarily make a breakthrough contribution that will significantly shift the field. As a result, I don't believe oral acceptance is warranted.

**Justification For Why Not Lower Score:**

Overall, the paper could be accepted as a poster as well. However, since this is a fast-moving and important field (especially in the areas of self-supervised learning) and also stands to reduce computational burden, I believe accepting as a spotlight is warranted.

**Metareview: Summary, Strengths And Weaknesses:**


 This paper takes a thorough look at ViT and Swin vision transformer architectures, and proposes a simpler architecture that is efficient (in terms of FLOPS/training time/etc.) while achieve strong results across a wide range of computer vision tasks. The focus is specifically on architecture designs that are easily compatible with masked image modeling (MIM) , given the importance of such self-supervised learning methods. There are a range of findings and architectural designs made in the paper, including the effectiveness network depth, relative position encoding (RPE), and hierarchical patch embeddings. Several more complex aspects of Swin are found not to be necessary, such as shifted window attention and additional stages. Overall, extensive experiments are conducted across supervised/self-supervised training regimes and several vision tasks. Importantly, other aspects such as FLOPs and training time are also shown, with strong results by the proposed method.

  The reviewers all appreciated the clear writing and analysis of these architectures, focus on masked image modeling, and thorough results. A number of weaknesses were highlighted, including veracity of the claims, missing experimental settings/ablations, and overall contributions of the findings since some of them were already present in the literature. These points, as well as some issues of clarity, were addressed by the authors and reviewers were all satisfied that their concerns have been addressed. Overall, this paper is a strong empirical study of vision transformers, especially in the self-supervised regime, with interesting insights and an effective new architecture that is simple, compatible with MIM, efficient, and effective at a range of vision tasks. As a result, it is a strong contribution to the community and I recommend acceptance.

**Note From Pc:**

if the above contains the word "oral" or "spotlight" please see: "oral" presentation means -> notable-top-5% and "spotlight" means -> notable-top-25%. As stated in our emails, we are disassociating presentation type from AC recommendations